# The Polyamine Analogue Ivospemin Increases Chemotherapeutic Efficacy in Murine Ovarian Cancer

**DOI:** 10.3390/biomedicines12061157

**Published:** 2024-05-23

**Authors:** Cassandra E. Holbert, Jackson R. Foley, Robert A. Casero, Tracy Murray Stewart

**Affiliations:** Sidney Kimmel Comprehensive Cancer Center at Johns Hopkins, Baltimore, MD 21231, USA; cholber2@jh.edu (C.E.H.); jfoley13@jh.edu (J.R.F.)

**Keywords:** polyamine, ivospemin, SBP-101, ovarian cancer, chemotherapy, polyamine analogue, platinum resistance

## Abstract

Polyamines are small polycationic alkylamines that are absolutely required for the continual growth and proliferation of cancer cells. The polyamine analogue ivospemin, also known as SBP-101, has shown efficacy in slowing pancreatic and ovarian tumor progression in vitro and in vivo and has demonstrated encouraging results in early pancreatic cancer clinical trials. We sought to determine if ivospemin was a viable treatment option for the under-served platinum-resistant ovarian cancer patient population by testing its efficacy in combination with commonly used chemotherapeutics. We treated four ovarian adenocarcinoma cell lines in vitro and found that each was sensitive to ivospemin regardless of cisplatin sensitivity. Next, we treated patients with ivospemin in combination with four commonly used chemotherapeutics and found that ivospemin increased the toxicity of each; however, only gemcitabine and topotecan combination treatments were more effective than ivospemin alone. Using the VDID8^+^ murine ovarian cancer model, we found that the addition of ivospemin to either topotecan or gemcitabine increased median survival over untreated animals alone, delayed tumor progression, and decreased the overall tumor burden. Our results indicate that the combination of ivospemin and chemotherapy is a worthwhile treatment option to further explore clinically in ovarian cancer.

## 1. Introduction

Polyamines are small, polycationic alkylamines that are essential for growth, proliferation, and survival in mammalian cells. Due to their positive charge at physiological pH, polyamines interact with negatively charged macromolecules to influence critical cellular processes including DNA structure and replication, gene expression, protein translation, proliferation, and apoptosis [1]. Cancer cells are acutely reliant upon elevated intracellular polyamine pools and dysregulate their polyamine metabolism as a means to support their continual proliferation. Dysregulation of polyamine metabolism can be accomplished by tumors through upregulation of polyamine biosynthesis, downregulation of polyamine catabolism, an increase in extracellular polyamine uptake, or a combination of the three [2,3,4]. Notably, upregulation of ornithine decarboxylase (ODC), a rate-limiting enzyme in polyamine biosynthesis, is correlated with increased polyamine pools in nearly every type of cancer [5,6]. Because of this commonality between tumor types, polyamine metabolism is a consequential target for potential cancer therapeutics.

Polyamine analogues are an encouraging therapeutic strategy aimed at exploiting the tight self-regulation of polyamine homeostasis in neoplastic cells. A promising class of these analogues involves the alkylation of the primary amine groups of spermine, the largest of the naturally occurring mammalian polyamines [7,8,9,10]. The traditional mechanism of action for polyamine analogues is to decrease intracellular polyamine levels through feedback inhibition. These compounds compete with endogenous polyamines for uptake and, upon intracellular accumulation, stimulate catabolism of higher-order polyamines and reduce polyamine biosynthesis [10,11,12,13]. By being sufficiently dissimilar to the natural polyamines, therapeutically relevant polyamine analogues are unable to support critical cellular functions, thereby starving cancer cells of necessary polyamines.

Several polyamine analogues, including bis(ethyl)norspermine (BENSpm/DENSpm) and diethylhomospermine (DEHSPM), have demonstrated efficacy against cancer both in vitro and in vivo. However, they elicited notable off-target effects and toxicities in early clinical trials [14,15,16,17,18]. Although it is likely that these off-target effects were a result of the dosing schemas and not the analogues themselves, the field has progressed by designing less-toxic derivatives of the first-generation polyamine analogues and improving dosing regimes [7]. Ivospemin (SBP-101) is a hydroxylated derivative of the spermine analogue DEHSPM (Figure 1). Previous studies have shown that ivospemin inhibits pancreatic and ovarian cancer both in vitro and in vivo [19,20]. Results from a multicenter phase 1a/b trial suggest that ivospemin is a tolerable and potentially advantageous addition to the standard of care (gemcitabine and nab-paclitaxel) in previously untreated metastatic pancreatic ductal adenocarcinoma (PDA) patients [21,22]. The combination of ivospemin and gemcitabine/nab-paclitaxel in metastatic PDA has progressed to the multi-center ASPIRE phase 2/3 clinical trial (NCT05254171) that is currently recruiting patients [23]. Mechanistically, our previous studies demonstrated that ivospemin decreases the viability of lung, pancreatic, and ovarian adenocarcinoma cell lines in vitro through modulation of polyamine biosynthetic and catabolic enzymes and reductions in intracellular polyamine pools [19]. Ivospemin treatment of a syngeneic ovarian cancer murine model significantly decreased the tumor burden and increased median survival [19].

Ovarian cancer was the third most commonly diagnosed gynecological cancer globally in 2020 and is predicted to rise in incidence over 40% by 2040 [24]. As over half of patients already have metastatic disease at diagnosis, ovarian cancer is the leading cause of gynecological cancer deaths, with a five-year relative survival for patients diagnosed with distant disease of approximately 30% [25]. First-line treatment consists of cytoreductive surgery with subsequent platinum-based chemotherapy; however, nearly 75% of patients will develop platinum-resistant tumors, with limited subsequent treatment options available [26,27,28]. As these patients represent a clear unmet therapeutic need, we investigated the potential of combining ivospemin treatment with chemotherapies commonly used in the treatment of platinum-resistant ovarian cancer.

## 2. Materials and Methods

### 2.1. Cell Lines, Culture Conditions, and Reagents

Ovarian adenocarcinoma cell line A2780 and its cisplatin-resistant derivative ACRP were maintained in RPMI 1640 containing 10% fetal bovine serum (GeminiBio, West Sacramento, CA, USA) [29]. Ovarian adenocarcinoma lines CaOV-3 and OV90 were maintained in DMEM supplemented with 10% fetal bovine serum. Human ovarian adenocarcinoma lines were purchased from the American Type Culture Collection (Manassas, VA, USA) and the European Collection of Cell Cultures (Salisbury, UK). ID8 mouse ovarian surface epithelial cells (MOSE) overexpressing VEGF and β-defensin (VDID8^+^) were obtained from Katherine Roby (University of Kansas, Lawrence, KS, USA) and maintained in RPMI 1640 supplemented with 10% fetal bovine serum. The polyamine analogue ivospemin (SBP-101) was obtained from Panbela Therapeutics, Inc. (Waconia, MN, USA). All chemotherapeutics were purchased commercially from Sigma-Aldrich (St. Louis, MO, USA).

### 2.2. Cell Viability Assays

Cells were seeded in triplicate wells per condition with a total cell count of 2 × 10^3^ cells per well of a 96-well plate and allowed to adhere overnight. Cells then received 100 μL of fresh medium with increasing concentrations of ivospemin. Following 96 h of ivospemin monotherapy, 20 μL of CellTiter-Blue reagent (Promega, Madison, WI, USA) was added, and following a 3 h incubation, the fluorescence was measured in black, clear-bottom plates at 560_Ex_/590_Em_ on a SpectraMax M5 (Molecular Devices, Sunnyvale, CA, USA). Wells containing medium alone were used as background controls. The combination chemotherapy/ivospemin experiment conditions were identical with the following exception: after 72 h of ivospemin exposure, cells received fresh medium with ivospemin and the appropriate chemotherapeutic. Following an additional 24 h (96 total hours of ivospemin, 24 h of chemotherapy), cell viability was determined using the CellTiter-Blue cell viability assay. Results are presented as percentages relative to untreated cells. Each bar is an average of at least three independent biological experiments (each experiment is noted by an individual point).

### 2.3. Intracellular Polyamine Pool Determination

Perchloric acid-extracted lysates from treated cells were labeled with dansyl chloride (Sigma-Aldrich, St. Louis, MO, USA) for detection using HPLC as previously described [30]. Polyamine concentrations were quantified relative to the total cellular protein in the lysates as determined by a Bradford assay (Bio-Rad Laboratories, Hercules, CA, USA) and are presented as the nmol polyamine concentration per mg of cellular protein [31].

### 2.4. Syngeneic Mouse Model

Female C57BL/6J wild-type mice (7–8 weeks old, The Jackson Laboratory, Bar Harbor, ME, USA) were housed at the Johns Hopkins Sidney Kimmel Comprehensive Cancer Center Animal Resources Core and cared for in accordance with the policies set forth by the Johns Hopkins University Animal Care and Use Committee. VEGF-β-defensin ID8 (VDID8^+^) syngeneic mouse ovarian surface epithelial cells (first developed by Dr. Katherine Roby) were injected intraperitoneally into C57BL/6 mice (350,000 cells/mouse) [32]. Treatment of the mice began three days after VDID8^+^ injection. Mice in the combination study were treated with the following dosing schedule: 24 mg/kg ivospemin 2qw × 3, alternating weeks; 30 mg/kg gemcitabine qw × 4; 1 mg/kg topotecan 3qw × 4. All drugs were administered intraperitoneally and are soluble in phosphate-buffered saline (PBS). Following the production of palpable ascites fluid or a 15% weight gain, animals were drained of their ascites fluid, which was measured as a marker for the tumor burden. The ascites fluid was drained into a 15 mL collection conical using an 18-gauge needle. Decisions for subsequent drains were determined by the same criteria. No animals were drained more than once a week, and animals who survived to a fourth drain were subsequently euthanized.

### 2.5. Statistical Analysis

All statistical testing was completed using GraphPad Prism software (v9.5.1, La Jolla, CA, USA). Dose responses of ovarian cells to ivospemin were plotted by a non-linear regression to determine IC_50_ values. Cell viability experiments were analyzed by one-way ANOVA and then individual comparisons were analyzed by two-sided Welch’s *t*-tests unless otherwise specified. Multiple testing correction was performed based on the Benjamini–Hochberg method. All data passed the Shapiro–Wilk test for normality. All mouse survival curves were analyzed using the log–rank (Mantel–Cox) test. Mouse ascites data were analyzed by one-way ANOVA and then individual comparisons were analyzed by two-sided Welch’s *t*-tests unless otherwise specified. Multiple testing correction was performed based on the Benjamini–Hochberg method. All data passed the Shapiro–Wilk test for normality. A *p*-value of <0.05 was considered statistically significant. *p*-value indications are as follows: * < 0.05; ** < 0.01; *** < 0.001; **** < 0.0001.

## 3. Results

### 3.1. Ivospemin Co-Treatment Increases Chemotherapy Toxicity in Ovarian Adenocarcinoma Cell Lines Regardless of Cisplatin Sensitivity

Cell lines (A2780, OV90, ACRP, and CaOV-3) with varying levels of cisplatin sensitivity (Appendix A) were treated for 96 h with concentrations of ivospemin ranging from 500 nM to 10 μM (Figure 2) [29,33,34,35]. All four lines, regardless of cisplatin sensitivity, responded to ivospemin treatment with IC_50_ values ranging between 0.95 μΜ and 3.24 μM (Table 1). All of these IC_50_ values are well below the previously published IC_50_s of pancreatic cancer cell lines (4.4 μM–8.4 μM), a tumor type already being evaluated clinically for ivospemin efficacy [19]. These four ovarian adenocarcinoma lines were then treated with various chemotherapeutic agents including gemcitabine (50 nM), topotecan (50 nM), paclitaxel (2 nM), and docetaxel (2 nM). As ivospemin use is of most interest clinically in patients with refractory disease, these agents represent standard-of-care chemotherapeutic options used with some regularity in patients with platinum-resistant ovarian cancer. Cells were treated with three increasing concentrations of each chemotherapy alone to determine an appropriate dose prior to combination studies.

Each ovarian adenocarcinoma line was treated with 2 μΜ of ivospemin in addition to each chemotherapeutic agent (Figure 3). The response to each chemotherapeutic agent was increased by ivospemin co-treatment in A2780 cells; however, only in combinations with gemcitabine or topotecan was this toxicity increase statistically significant over that of either agent alone (*p*-values of 0.045 and 0.0023, respectively, compared to ivospemin treatment) (Figure 3A). Similarly, ivospemin co-treatment increased the toxicity of each chemotherapeutic agent in ACRP cells, the cisplatin-resistant derivative of A2780, with the reciprocal treatment of gemcitabine and topotecan resulting in greater toxicity than ivospemin alone (*p*-values of 0.02 and 0.0006, respectively) (Figure 3B). Ivospemin did not increase the response to paclitaxel or docetaxel in OV90 cells, but it did increase the cellular response to both gemcitabine and topotecan (*p*-values of 0.01 and 0.002, respectively) (Figure 3C). Similarly, ivospemin treatment only significantly increased the response to gemcitabine and topotecan in CaOV-3 cells (*p*-values of 0.005 and 0.004, respectively) (Figure 3D). The addition of ivospemin increases the reduction in cellular viability in each evaluated chemotherapeutic agent. In each cell line, ivospemin increases both topotecan and gemcitabine’s toxicity in a statistically significant manner (Table 2). While cell viability was decreased in all cell lines following the addition of ivospemin to either paclitaxel or docetaxel, only a subset of cell lines decreased viability in a statistically significant manner (Table 2). Overall, gemcitabine and topotecan comparably increased the response to ivospemin in all four ovarian adenocarcinoma lines tested, while paclitaxel and docetaxel treatment consistently added little to no benefit to ivospemin monotherapy in any of the tested cell lines. However, ivospemin addition improved the response to each of the four tested chemotherapeutic agents. Ivospemin decreased intracellular polyamine levels in all treated cell lines (Appendix A). The addition of chemotherapy treatment to ivospemin-treated cells did not impact polyamine depletion, suggesting that the additive benefit of ivospemin is independent of each chemotherapeutic’s mechanism of action.

### 3.2. Ivospemin Treatment in Combination with Chemotherapy Increases Survival in a Murine Ovarian Adenocarcinoma Model Compared to Chemotherapy Alone

We previously demonstrated that ivospemin reduced the tumor burden and polyamine levels in the ascites while extending survival in a murine ovarian model in vivo [19]. To further evaluate the potential benefit of ivospemin addition to gemcitabine and topotecan in vivo, female C57Bl/6J mice were injected with 350,000 VDID8^+^ syngeneic ovarian epithelial cancer cells and subsequently treated with the appropriate drugs. Similar to our monotherapy study [19], ivospemin alone produced a 20% increase in median survival when compared to control animals (Figure 4). Neither gemcitabine nor topotecan produced a statistically significant survival benefit as a monotherapy (Table 3). The addition of ivospemin to topotecan increased survival by 28% compared to topotecan monotherapy, and the addition of ivospemin to gemcitabine increased survival by 24% compared to gemcitabine alone (Figure 4). Treatment with either double combination resulted in an approximately 45% increase in median survival compared to untreated animals.

### 3.3. Ivospemin Increases Chemotherapeutic Efficacy by Delaying Disease Onset and Decreasing Overall Tumor Burden in a Murine Ovarian Adenocarcinoma Model

There is an average of 37.5 days between VDID8^+^ cell injection and measurable ascites formation in untreated animals as determined by either 15% weight gain or visible abdominal swelling (Figure 5A). Ivospemin alone does not significantly affect the time to ascites formation, with an average time of 39.5 days. Gemcitabine and topotecan each modestly delayed ascites formation, with average times of 41.7 and 43.4 days, respectively. The addition of ivospemin to gemcitabine extended the time to ascites formation to 46.9 days, an increase of 13%, while the ivospemin and topotecan combination mice exhibited ascites 50 days post VDID8^+^ injection, a 15% increase (Figure 5A).

Untreated mice produced an average of 3 mL of ascites fluid at their first drain (Figure 5B). Ivospemin monotherapy appeared to decrease overall ascites production (a measure of tumor burden), with an average ascites volume at first drain of 1.15 mL, though this decrease remains a trend as it did not reach the level of statistical significance (*p*-value = 0.0541). Similarly, gemcitabine and topotecan monotherapies had no impact on ascites formation, with average first drain volumes of 2.03 and 3.19 mL, respectively (Figure 5B). Although the addition of ivospemin to either monotherapy trended toward decreasing the average ascites fluid but did not reach statistical significance at first drain, ivospemin maintained its influence on ascites volume through subsequent drains while the monotherapy volumes increased (Figure 5B). Ivospemin-treated mice produced 50% less ascites than untreated mice at their second drain (*p*-value = 0.0027) (Figure 5C), and its addition to the chemotherapies also reduced ascites volume: adding ivospemin to gemcitabine or topotecan reduced ascites formation by 46% (*p*-value = 0.006) and 53% (*p*-value = <0.0001), respectively (Figure 5C).

## 4. Discussion

Ovarian cancer is the leading cause of death amongst gynecologic malignancies, with many deaths being attributable to the development of platinum-resistant tumors [24,25]. The main therapeutic strategy for platinum-resistant ovarian cancer is systemic chemotherapy, such as gemcitabine or liposomal doxorubicin, which only has an effective rate of approximately 10–30% [26,27]. As such, the discovery of new therapeutic options and combinations is essential to overcome the unmet therapeutic needs of the platinum-resistant patient population.

Ivospemin is a spermine analogue that has shown promising efficacy against pancreatic and ovarian cancers both in vitro and in vivo [19,20]. Through suppression of polyamine biosynthesis and upregulation of polyamine catabolism, ivospemin depletes essential polyamine pools from continuously growing cancer cells in vitro [19]. Additionally, ivospemin has been shown to be tolerable and potentially advantageous in a phase 1 clinical trial in pancreatic ductal adenocarcinoma [21]. Here, we evaluated the efficacy of the novel combination of ivospemin with a variety of chemotherapeutic agents used in platinum-refractory disease. Our in vitro studies determined that ivospemin reduces cell viability in four tested ovarian adenocarcinoma cell lines and increases the effect of chemotherapeutics including gemcitabine, topotecan, paclitaxel, and docetaxel regardless of the cell line’s sensitivity to cisplatin (Figure 2 and Figure 3). While all four human lines have ivospemin IC_50_ values well below those previously documented in other cancer types, there is variation in the sensitivity of each cell line that appears to be independent of cisplatin sensitivity (Figure 2) [19]. It is likely that the mutational landscape of each cell line plays a role in the level of ivospemin sensitivity, and future studies will evaluate transcriptomic changes following ivospemin treatment across multiple genomic backgrounds to determine genetic factors that may influence sensitivity. While ivospemin increased the response of each of the four chemotherapeutic agents tested, only the combination of ivospemin with gemcitabine or topotecan in vitro regularly produced a significant decrease in viability compared to ivospemin monotherapy. Importantly, ivospemin efficacy appears to be independent of cisplatin sensitivity. These data suggest that ivospemin may be a valuable addition to standard chemotherapy, but that some cancers may respond as well to ivospemin monotherapy as to the combination treatment.

Utilizing a murine model of ovarian cancer, we have shown that ivospemin monotherapy increases median survival by 20% (Figure 4). Ivospemin treatment also increases the survival of gemcitabine- and topotecan-treated mice by approximately 25%. The addition of ivospemin to either chemotherapeutic modestly increases the time from cancer cell injection to ascites formation (Figure 5A). While ivospemin treatment delays ascites formation in combination arms, this alone does not appear to fully account for the observed increase in survival. Ivospemin alone does not delay ascites onset in a statistically significant manner; however, monotherapy still produces a clear increase in median survival (Figure 4 and Figure 5A). Notably, ivospemin treatment decreases the overall tumor burden (as measured by ascites volume) in both monotherapy and combination therapy arms (Figure 5B,C). Ivospemin monotherapy reduces ascites volume by more than 50%, and combination therapy exhibits a similar effect regardless of the chemotherapeutic agent being evaluated (Figure 5B). This reduction is sustained throughout treatment and becomes more substantial with increasing time of treatment (Figure 5C). Importantly, ivospemin reduces ascites volume to a similar level whether dosed as a single or combination agent, indicating that the decrease in tumor burden can be mostly, if not completely, attributed to ivospemin treatment alone.

Taken together, our data further support the potential of ivospemin as a clinical agent in ovarian cancer. Previous studies have demonstrated the safety profile of ivospemin in humans, and early results indicate that ivospemin may provide a survival benefit over standard-of-care chemotherapy in pancreatic cancer. Previous work has shown that platinum-based drugs, such as cisplatin and oxaplatin, can influence polyamine metabolism, and the polyamine analogue N1,N11-diethylnorspermine (DENSpm) has exhibited combinatorial effects with platinum-based drugs in ovarian adenocarcinoma cell lines [36,37,38]. It is possible that ivospemin may also exhibit a combinatorial effect with platinum-based therapies, and this is an important future area of study. Our data, however, focus on the underserved platinum-resistant patient population and provide evidence of ivospemin’s ability to be combined with chemotherapy across an additional tumor type. Most importantly, our data support the conclusion that ivospemin may be of clinical benefit in addition to standard-of-care agents, providing a new therapeutic option for patients with platinum-resistant ovarian tumors.

## Figures and Tables

**Figure 1 biomedicines-12-01157-f001:**
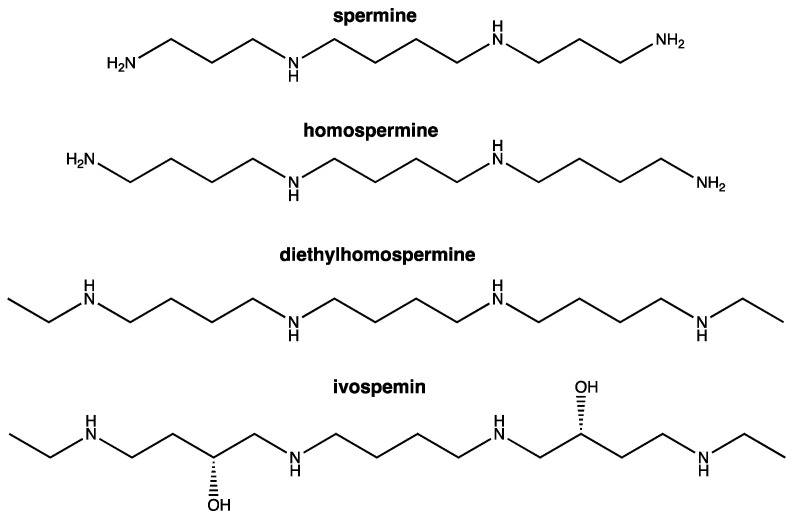
Structures of spermine analogue derivatives including ivospemin. Homospermine is an analogue of spermine produced by the symmetrical addition of two methylene groups. The analogue is further derived to diethylhomospermine by the addition of ethyl groups to either end of the molecule. Ivospemin is a hydroxylated derivative of diethylhomospermine.

**Figure 2 biomedicines-12-01157-f002:**
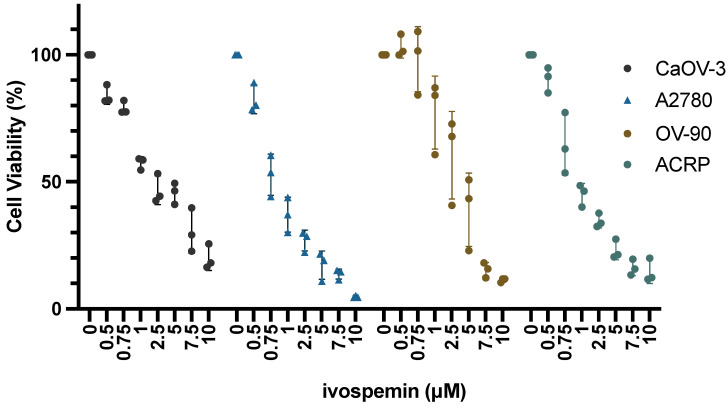
Ivospemin treatment reduces cell viability in human ovarian adenocarcinoma cells in vitro regardless of cisplatin sensitivity. Ovarian adenocarcinoma cell lines were treated for 96 h with increasing concentrations of ivospemin ranging from 500 nM to 10 μM. Ivospemin treatment decreased cell viability in all four cell lines tested. These four lines represent varying levels of cisplatin sensitivity, with CaOV-3 being the most sensitive and ACRP being the least sensitive. Ivospemin and cisplatin IC_50_ values are listed in Table 1.

**Figure 3 biomedicines-12-01157-f003:**
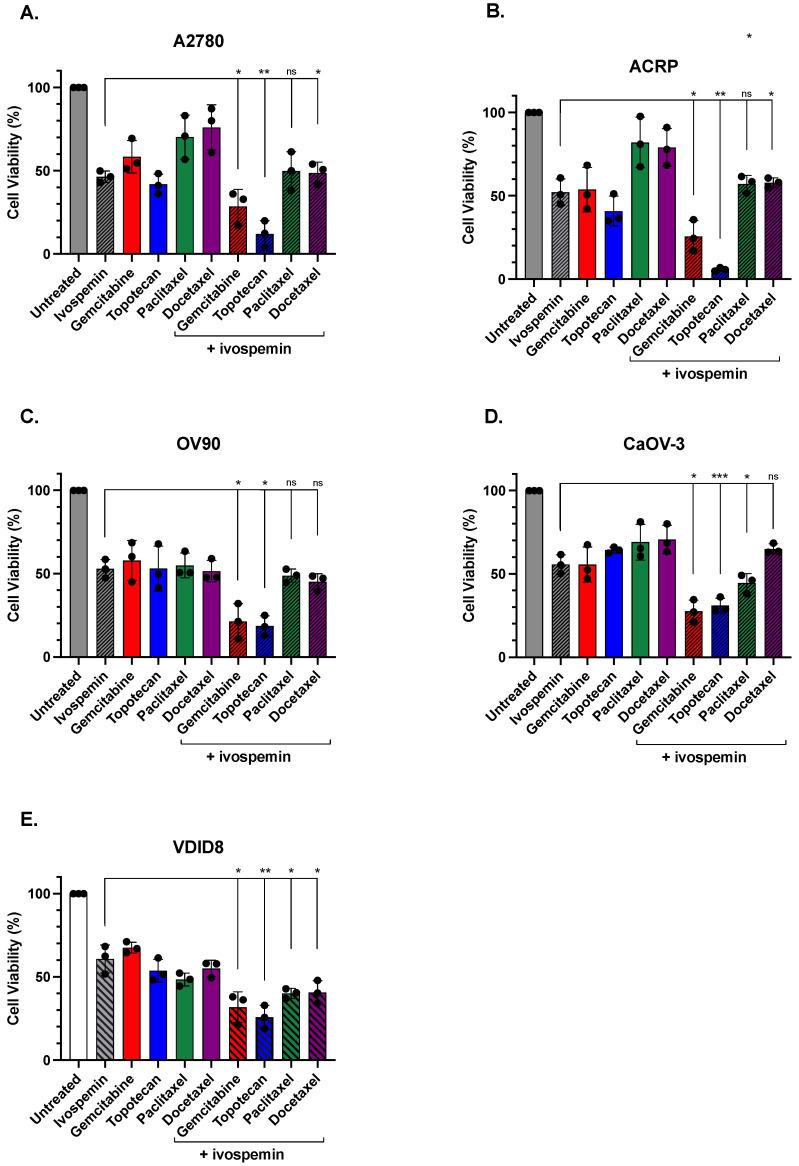
Ivospemin increases the response to gemcitabine and topotecan in human and mouse ovarian cancer cell lines. Four human ovarian adenocarcinoma cell lines and one murine ovarian cancer cell line were treated with 2 μM of ivospemin for 96 h, with the inclusion of chemotherapeutic agents for the last 24 h at the following concentrations: gemcitabine: 50 nM; topotecan: 50 nM; paclitaxel: 2 nM; docetaxel 2 nM. Ivospemin increased the toxicity of all four chemotherapeutics in both A2780 and ACRP cell lines (**A**,**B**), but combinations with gemcitabine and topotecan most notably exceeded the effect of ivospemin alone. Ivospemin similarly increased the toxicity of gemcitabine and topotecan in the remaining human ovarian cancer cell lines, OV90 (**C**) and CaOV-3 (**D**), with limited effect on either taxane drug. Ivospemin increased the toxicity of all four chemotherapeutics in the murine VDID8^+^ cell line (**E**). *p*-value indications are as follows: ns > 0.05; * < 0.05; ** < 0.01; *** < 0.001.

**Figure 4 biomedicines-12-01157-f004:**
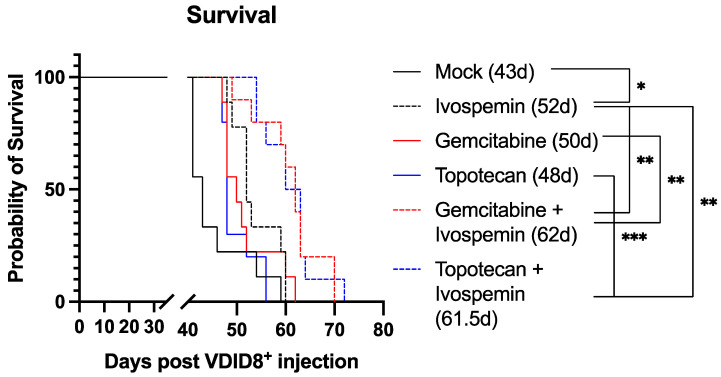
Ivospemin increases survival over chemotherapy alone in the VDID8^+^ model of ovarian cancer. Female C57Bl/6J mice were injected with 350,000 VEGF^+^, defensin^+^ ID8 cells per mouse. Treatment with all three drugs began on day 3 post-injection at the following doses: ivospemin (24 mg/kg 2qw × 3, alternating weeks); gemcitabine (30 mg/kg qw × 4); topotecan (1 mg/kg 3qw × 4). Ivospemin monotherapy increased median survival from 43 days to 52 days post-injection. Both combination therapies increased median survival to approximately 62 days post-injection (~45% increase). *n* = 10/group. *p*-value indications are as follows: ns > 0.05; * < 0.05; ** < 0.01; *** < 0.001.

**Figure 5 biomedicines-12-01157-f005:**
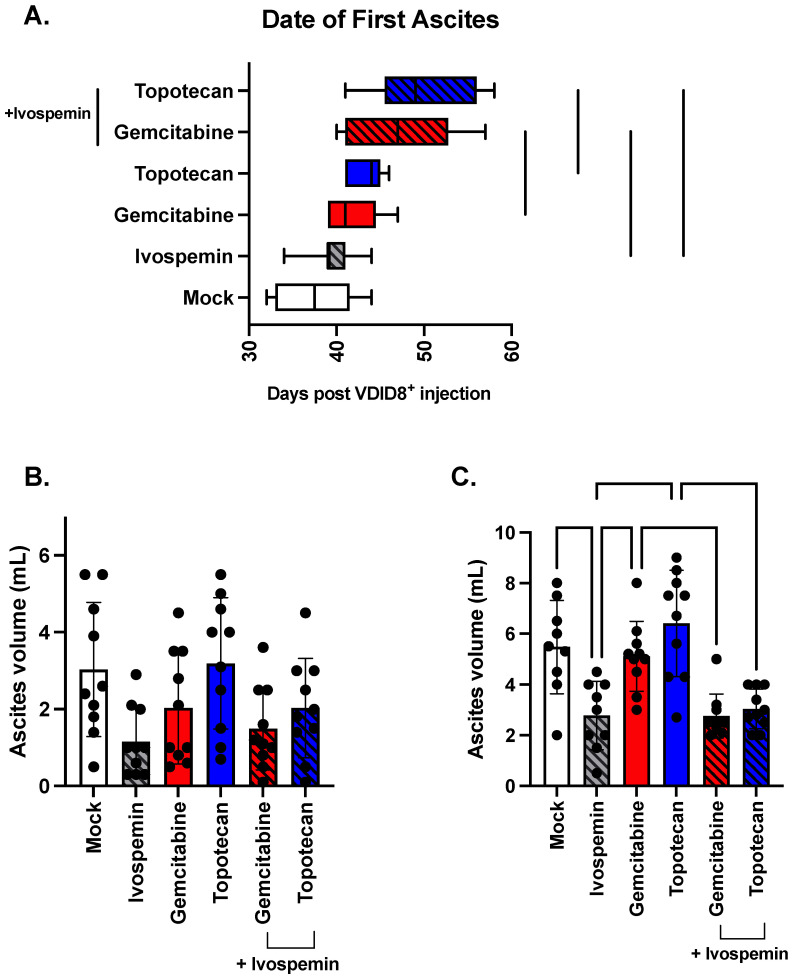
Ivospemin improves the survival benefit of chemotherapy by delaying disease onset and decreasing the tumor burden. While ivospemin alone did not influence time to ascites formation, the addition of ivospemin to either gemcitabine or topotecan monotherapy increased the number of days between cell injection and ascites formation (**A**). Ivospemin treatment trends towards a decrease in ascites volume at the first drain (**B**) either as a monotherapy or in combination with chemotherapy. Chemotherapy had no effect on ascites volume. This decreased ascites volume following ivospemin treatment was sustained and amplified through the second drain (**C**), resulting in a reduction in the tumor burden of approximately 50%.

**Table 1 biomedicines-12-01157-t001:** IC_50_ values following ivospemin treatment in human ovarian adenocarcinoma cell lines.

Cell Line	Cisplatin 48 h IC_50_	Ivospemin 96 h IC_50_
CaOV-3	4.24 μM	2.56 μM
A2780	5.21 μM	0.95 μM
OV90	14.13 μM	3.24 μM
ACRP	28.42 μM	1.41 μM

**Table 2 biomedicines-12-01157-t002:** Individual comparisons of ivospemin combination treatment vs. chemotherapeutic monotherapy.

Comparison	*p*-Value	Significance
Gemcitabine (A2780)	0.021	*
Gemcitabine (ACRP)	0.041	*
Gemcitabine (CaOV-3)	0.018	*
Gemcitabine (OV90)	0.016	*
Gemcitabine (VDID8)	0.016	*
Topotecan (A2780)	0.007	**
Topotecan (ACRP)	0.003	**
Topotecan (CaOV-3)	0.0003	***
Topotecan (OV90)	0.016	*
Topotecan (VDID8)	0.005	**
Paclitaxel (A2780)	0.114	ns
Paclitaxel (ACRP)	0.054	ns
Paclitaxel (CaOV-3)	0.025	*
Paclitaxel (OV90)	0.281	ns
Paclitaxel (VDID8)	0.016	*
Docetaxel (A2780)	0.034	*
Docetaxel (ACRP)	0.035	*
Docetaxel (CaOV-3)	0.338	ns
Docetaxel (OV90)	0.243	ns
Docetaxel (VDID8)	0.033	*

*p*-value indications are as follows: ns > 0.05; * < 0.05; ** < 0.01; *** < 0.001.

**Table 3 biomedicines-12-01157-t003:** Individual comparisons of median survival. Gemcitabine (gem), ivospemin (ivo), and topotecan (topo).

Comparison	*p*-Value	Significance
Mock vs. Ivo	0.026	*
Mock vs. Gem	0.053	ns
Mock vs. Topo	0.170	ns
Ivo vs. Gem	0.559	ns
Ivo vs. Topo	0.037	*
Ivo vs. Gem + Ivo	0.004	**
Ivo vs. Topo + Ivo	0.003	**
Gem vs. Gem + Ivo	0.004	**
Topo vs. Topo + Ivo	0.0002	***

*p*-value indications are as follows: ns > 0.05; * < 0.05; ** < 0.01; *** < 0.001.

## Data Availability

Data supporting results are contained within this article.

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
