# Peer review of "The Polyamine Analogue Ivospemin Increases Chemotherapeutic Efficacy in Murine Ovarian Cancer"

_biomedicines, 2024, doi:10.3390/biomedicines12061157_

Round 1
Reviewer 1 Report
Comments and Suggestions for Authors
The manuscript by Holbert et al, tests the efficacy of ivospemin, an inhibitor of the polyamine biosynthesis by acting as an analogue of spermine. Since the efficacy of various polyamine inhibitors/analogues and ivospemin has been demonstrated in various cancers including ovarian, the authors have focused here on testing the efficacy of ivospemin on chemo-resistant ovarian cancer models. The authors have used standard cell lines and techniques to in vitro and in vivo to demonstrate convincingly that ivospermin is effective. However, no data is presented on its effect on polyamine metabolism. Do the polyamine levels decrease? Are any of the enzyme's expression impacted. Does the sensitivity of the cell line depend on the basal levels of polyamines or spermine specifically, especially as the sensitive OV90 shows a higher IC50 compared to the resistant cell lines. this would be a key question to address prior to ivospemin considered to be more potent for chemoresistant cancer cells.
Author Response
The manuscript by Holbert et al, tests the efficacy of ivospemin, an inhibitor of the polyamine biosynthesis by acting as an analogue of spermine. Since the efficacy of various polyamine inhibitors/analogues and ivospemin has been demonstrated in various cancers including ovarian, the authors have focused here on testing the efficacy of ivospemin on chemo-resistant ovarian cancer models. The authors have used standard cell lines and techniques to in vitro and in vivo to demonstrate convincingly that ivospemin is effective. However, no data is presented on its effect on polyamine metabolism. Do the polyamine levels decrease? Are any of the enzyme's expression impacted. We have previously shown that ivospemin decreases polyamine content through upregulation of SSAT and downregulation of ODC (See lines 61-65 and doi: 10.3390/ijms23126798). The mechanisms of all four chemotherapeutic agents have been well documented. As these mechanisms do not overlap with the understanding of ivospemin, and the response is additive, we did not originally evaluate polyamines and enzymes in these particular studies. We have subsequently verified that ivospemin treatment depletes polyamine levels in human ovarian adenocarcinoma lines (Supplemental Figure 2) and that the addition of chemotherapy does not impact this depletion. As such, we did not complete enzymatic activity work as we have previously documented the mechanism by which ivospemin depletes polyamine levels in vitro. We have added text indicating such in lines 214-218. Does the sensitivity of the cell line depend on the basal levels of polyamines or spermine specifically, especially as the sensitive OV90 shows a higher IC50 compared to the resistant cell lines. this would be a key question to address prior to ivospemin considered to be more potent for chemoresistant cancer cells. The sensitivity column on Table 1 is referring to cisplatin sensitivity not ivospemin sensitivity. The legend and labeling has been improved for clarity. The IC50s in response to ivospemin for the ovarian lines range from 0.95 μM to 3.24 μM, notably less than the IC50 values for other cell lines (such as pancreatic) that are already being evaluated clinically. We have added the following statement to underline this point in the text “These IC50 values are well below the previously published IC50s of pancreatic cancer cell lines (4.4 μM – 8.4 μM), a tumor type already being evaluated clinically for ivospemin efficacy.” Our goal with these studies is to demonstrate sensitivity to ivospemin that is independent of cisplatin response. We are not suggesting that ivospemin will be more potent in platinum-drug-resistant cancers.
Reviewer 2 Report
Comments and Suggestions for Authors
No problem with English.
Author Response
Overall Comments: The manuscript lacks sufficient discussion on the use of cisplatin and paclitaxel in all cell lines studied. Paclitaxel was evaluated in four human cell lines but not in VDID8+, and there is a lack of IC50 data for cisplatin across all cell lines. IC50 data for cisplatin has been added to Table 1 and the data are shown in Supplementary Figure 1. As paclitaxel failed to provide significant benefit over ivospemin alone, it was not tested in the murine model. The author mentions that these four human cell lines are either sensitive or resistant, but the rationale behind this categorization is not well-explained. We have reformulated our categorization as varying levels of cisplatin sensitivity based on the IC50 data we have included. We have also included the following references to our cell lines and their cisplatin sensitivities (Ref #29, 33-35) While the inclusion of topotecan and gemcitabine as chemotherapy agents is uncommon but not unheard of in treating advanced ovarian cancer, the manuscript should primarily focus on platinum-based drugs (cisplatin, carboplatin) in combination with taxanes along with ivospemin. The purpose of this study was to focus on the unmet need for patients who have already developed platinum-resistant disease. The inclusion of gemcitabine, topotecan and the taxanes were decided on after consultation with physicians interested in developing a phase 1 ivospemin trial. The enrolled population for a phase 1 will be patients with platinum-resistant disease explaining our omission of cisplatin. We agree that the combination of platinum-based drugs with ivospemin is of interest and warrants subsequent studies, however these experiments are beyond the current scope of this manuscript. We have included this point in our discussion and have cited the following additional studies:
- Varma, R.; Hector, S.; Greco, W.R.; Clark, K.; Hawthorn, L.; Porter, C.; Pendyala, L. Platinum drug effects on the expression of genes in the polyamine pathway: time-course and concentration-effect analysis based on Affymetrix gene expression profiling of A2780 ovarian carcinoma cells. Cancer Chemother Pharmacol 2007, 59, 711-723, doi:10.1007/s00280-006-0325-3.
- Hector, S.; Porter, C.W.; Kramer, D.L.; Clark, K.; Prey, J.; Kisiel, N.; Diegelman, P.; Chen, Y.; Pendyala, L. Polyamine catabolism in platinum drug action: Interactions between oxaliplatin and the polyamine analogue N1,N11-diethylnorspermine at the level of spermidine/spermine N1-acetyltransferase. Mol Cancer Ther 2004, 3, 813-822, doi:3/7/813 [pii].
- Tummala, R.; Diegelman, P.; Hector, S.; Kramer, D.L.; Clark, K.; Zagst, P.; Fetterly, G.; Porter, C.W.; Pendyala, L. Combination effects of platinum drugs and N1, N11 diethylnorspermine on spermidine/spermine N1-acetyltransferase, polyamines and growth inhibition in A2780 human ovarian carcinoma cells and their oxaliplatin and cisplatin-resistant variants. Cancer Chemother Pharmacol 2011, 67, 401-414, doi:10.1007/s00280-010-1334-9.
Although the use of ivospemin in ovarian cancer is novel, it would strengthen the study to analyze alternative compensatory pathways using genomic, transcriptomic, or proteomic platforms to justify the use of drug combinations. Ongoing studies evaluating transcriptomic responses to ivospemin in ovarian cancer are underway to aid in informing new drug combinations but are beyond the scope of the current manuscript. The mechanistic action of ivospemin is not adequately addressed, with only observational data presented. The mechanism of action of ivospemin was reported by our group in a previous publication (doi: 10.3390/ijms23126798), however we have added more detailed reporting of the mechanism in our introductory section. While the use of the murine model is appropriate, the rationale for using the VDID8+ cell line is not well-supported. Although VDID8+ is Trp53 WT, its molecular signature differs from high-grade serous ovarian cancer (HGSOV) in humans. However, it's reasonable to suggest that a small portion of HGSOV cases may have TP53 WT status. The lack of in vitro drug sensitivity testing for VDID8+ cells makes it difficult to interpret the drug response in the in vivo model. We chose to use the VDID8+ model as it was the model previously used to evaluate ivospemin’s efficacy in reducing polyamine levels and tumor burden, while extending survival in vivo (doi: 10.3390/ijms23126798). Recognizing the limitations of the VDID8+ model, particularly in regard to its genetic status, we have recently started evaluating ivospemin in additional syngeneic ovarian models with more representational mutation status, however these models were not available to us during the completion of this study. Future studies, including the ongoing transcriptomic study above, will use the new model system. We have added a statement in our discussion/conclusion to indicate such. We have also added VDID8+ in vitro drug sensitivities to Figure 3.
Specific Comments:
- The title is misleading as the manuscript includes four human cell lines and one murine cell line.
While the preliminary in vitro studies in the manuscript focus on human cell lines, the primary experiments indicate that ivospemin increases chemotherapeutic efficacy in vivo. As the in vivo studies are murine we feel our title is appropriate.
- Line 67: Is ovarian cancer the third most commonly diagnosed cancer in the US or globally?
The data are in reference to global cancer rates. We have added clarification to the text.
Lines 84-87: Where were the cell lines purchased from, and were they authenticated? It would be beneficial to include the TP53 status of these cell lines. We have added information regarding purchasing of the cell lines to the methods text. ACRP and its parent A2780 are p53 WT while both CaOV-3 and OV90 are p53 mutant (Gln136Ter and Ser215Arg, respectively). While it is likely that each cell line’s genetic makeup influences its level of ivospemin sensitivity, all four cell lines regardless of p53 status are more responsive than cell lines of previously tested tumor types. We have added the following statement to the discussion to address this “It is likely that the mutational landscape of each cell line plays a role in the level of ivospemin sensitivity, and future studies will evaluate transcriptomic changes following ivospemin treatment across multiple genomic backgrounds to determine genetic factors that may influence sensitivity.”
- How was ascitic fluid draining performed? We have added this information to the methods section.
- Lines 138-319: Without actual IC50 data from experiments, it's challenging to determine whether these cells are truly cisplatin-sensitive or resistant. We have reframed this as “varying levels of cisplatin sensitivity” and have included IC50 for the four human cell lines used and additional literature references as stated above.
- There is no in vitro data available for VDID8+ cells with any of the drugs used. We have added in vitro data from VDID8+ to Figure 3.
- Figure 3 should include cisplatin-treated cells. We have added cisplatin sensitivities and IC50s as stated above. The focus on Figure 3 is on ivospemin in combination with non-platinum chemotherapeutics.
- In Figure 4, how many mice were included per group? Ten mice per group were used. We have added the information to the figure legend.
- How long does it take for ascites to resume after the first drainage in both the control and treatment groups? Similar to the phenotype in humans, ascites formation resumes quickly after draining. As stated in the methods, mice are typically drained once per week after they have produced measurable ascites regardless of treatment group.
- What do tumor nodules in the peritoneal cavity look like, and were histological analyses performed?
Intraperitoneal injection of VDID8+ cells allows for peritoneal spread of cells and ascites formation but does not typically induce solid tumor formation. Occasionally in this model, mice will form small nodules that tend to be diaphragmatic or on the peritoneal wall. A benefit of the VDID8+ model is that tumor burden has been shown to be directly correlated with hemorrhagic ascites formation (doi: 10.1016/s0002-9440(10)64505-1 and 10.1038/nm1097). As such, necropsy and histological analysis was not performed on these mice.
- The study lacks mechanistic insights into the action of the drugs tested, including single and combination therapies. The absence of western blot data to support drug inhibition is noted.
The mechanism of action of ivospemin was reported by our group in a previous publication (doi: 10.3390/ijms23126798) and the mechanism of action of the chemotherapeutics used is already well documented. Our combinatorial data indicate an additive benefit suggesting there is not synergistic mechanistic work to complete. Most ODC regulation occurs post-translationally. Our lab has previously shown that ivospemin inhibits ODC activity in numerous cell types, therefore western blotting does not provide additional information.
Round 2
Reviewer 2 Report
Comments and Suggestions for Authors
Thank you for your effort in revising the manuscript. You have done a fantastic job